# MUSAK: A Multi-Scale Space Kinematic Method for Drone Detection

**Sunxiangyu Liu [1,2], Guitao Li [2], Yafeng Zhan [1] and Peng Gao [3,*]**

[1] Beijing National Research Center for Information Science and Technology (BNRist), Tsinghua University, Beijing 100084, China; lsxy14@mails.tsinghua.edu.cn (S.L.); zhanyf@tsinghua.edu.cn (Y.Z.)
[2] School of Aerospace Engineering, Tsinghua University, Beijing 100084, China; ligt@tsinghua.edu.cn
[3] School of Mechatronical Engineering, Beijing Institute of Technology, Beijing 100081, China
[*] Correspondence: gaopeng1982@pku.edu.cn

**Abstract:** Accurate and robust drone detection is an important and challenging task. However, on this issue, previous research, whether based on appearance or motion features, has not yet provided a satisfactory solution, especially under a complex background. To this end, the present work proposes a motion-based method termed the Multi-Scale Space Kinematic detection method (MUSAK). It fully leverages the motion patterns by extracting 3D, pseudo 3D and 2D kinematic parameters at three scale spaces according to the keypoints quality and builds three Gated Recurrent Unit (GRU)-based detection branches for drone recognition. The MUSAK method is evaluated on a hybrid dataset named multiscale UAV dataset (MUD), consisting of public datasets and self-collected data with motion labels. The experimental results show that MUSAK improves the performance by a large margin, a 95% increase in average precision (AP), compared with the previous state-of-the-art (SOTA) motion-based methods, and the hybrid MUSAK method, which integrates with the appearance-based method Faster Region-based Convolutional Neural Network (Faster R-CNN), achieves a new SOTA performance on AP metrics (AP, APM, and APS).

**Keywords:** drone detection; motion-based; kinematic; multi-scale space

## 1. Introduction

UAVs (Unmanned Aircraft Vehicles), or drones, are currently being widely utilized in civilian and military applications, such as surveillance, rescue, surveying and delivery, for their features of "LSS" (low altitude, slow speed, and small size). However, as a result of such characteristics, UAVs are also hard to detect, and therefore may cause serious threat to military and social security, especially for airplanes when landing or taking off. For example, Frankfurt Airport temporarily closed in March 2019 due to two hovering drones nearby, and caused approximately 60 flight cancellations [1]. Hence, an accurate, long and large range UAVs detection method is urgently required for now and the future.

Recent approaches for detecting UAVs in images are always based on computer vision (CV) methods [2–5], which can be roughly classified into three categories: those based on appearance, on motion information across frames, and the hybrid. Appearance-based methods rely on specially designed neural network (NN) frameworks, such as Faster R-CNN [6], You-Only-Look-Once (YOLO)v3 [7], Single Shot MultiBox Detector (SSD) [8] and Cascade R-CNN [5]. They have been proven to be powerful under complex lighting or backgrounds for some tasks. However, their limitation is that the targets are required to be relatively large and clear in vision [9,10], which is often not the case in real-world scenes for drone detection. Motion-based methods mainly rely on optical flow [11–16] or motion modeling of foreground [17–19]. These methods are more robust when the target objects are tiny or blurry in images, but they are more often employed for region proposal or distinguishing moving objects from static backgrounds, rather than detecting. Hybrid methods, combining both appearance and motion information, may add extra structures

or restraints, such as motion restrictions [8,20], or inter-frame coherence [21,22] to basic detection architectures (typically appearance-based neural network backbones). In short, previous approaches have exploited different characteristics of drones and significantly improved detection performance. However, the common shortage of those methods is they only consider the problem of drone detection under a relatively clean background with one or two targets from few distractors. Moreover, rare work has been done on exploiting motion features for object recognition, let alone for drones.

Compared with other conventional object detection problems, the drone detection problem poses the following unique challenges: First, the object to be detected may appear in any area of the frame and move towards any directions. Second, the background of an object is always complex and changes fast in city scenes. Third, disturbances of UAV-like targets, such as birds, kites or pedestrians are commonly seen. Forth, target has usually less than 200 pixels in the captured image, and its appearance is various too, which results in severe performance degeneration of CNN-based methods.

To address the problem, the present work introduces a novel motion-based method named the multi-scale space kinematic detection method (MUSAK). It relies on recovery of object motion behavior, which is inspired by the observation that different objects have different motion behaviors. The MUSAK method detects drones from uncontrolled backgrounds by exploiting multiscale kinematic parameters extracted from input videos. The kinematic parameters here consist of the translation parameters (translation velocity and acceleration) and rotation parameters (angular velocity and acceleration).

The structure of the MUSAK method is shown in Figure 1. The pipeline starts from the extraction and tracking of ROIs (Regions of Interest). Then, based on the number and quality of the keypoints in each ROI, the process goes into one of the three compatible scale spaces to further extract the kinematic parameters. Afterwards, the extracted time-series motion parameters are fed to the corresponding GRU classifiers (they are separately trained on a customized UAV database comprised of several public datasets and the homemade multiscale UAV dataset (MUD)), to output the motion recognition results. Different from the previous detection methods, MUSAK extracts the motion patterns by exploiting the kinematic parameters at different scales, which enables it to enlarge the interclass differences to a great extent.

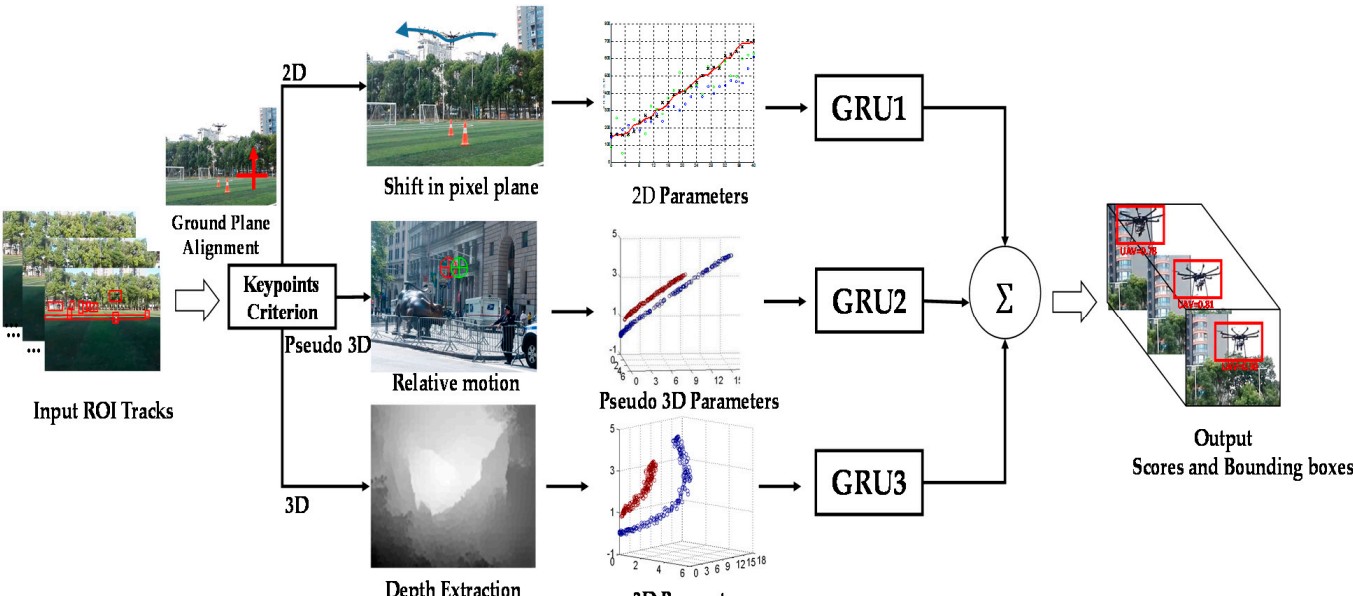

**Figure 1.** Pipeline of the MUSAK method.

The experiments suggest that MUSAK achieves the state-of-the-art detection accuracy for UAVs compared with the existing methods. We also carry out the adaptivity and significance analyses for MUSAK.

The main contributions can be summarized as follows:

1. For the first time, this work proposes a motion-based method that uses rigid body kinematic parameter combinations for detecting objects.
2. The proposed MUSAK introduces three scale spaces to describe the different state of the kinematic parameters of objects. Particularly, it uses a pixel amount of the object as a relative depth to construct the pseudo 3D space.
3. A new drone datasets MUD is established. It is comprised of several public databases and newly added data in real-world scenes with motion-related information labels.

The remainder of this paper is organized as follows. We give an overview on related work in Section 2. In Section 3, we describe each part of the proposed MUSAK method. The introduce of datasets and experimental results are shown in Section 4. Further discussion is stated in Sections 5 and 6 concludes our paper in short.

## 2. Related Work

Methods for detecting drones with CV methods can be classified into three categories: those relying on appearance features, on motion information across frames, and the hybrid. In this section, we will briefly review those studies as the foundation of our work.

### 2.1. Appearance-Based Methods

Targeting the LSS features of UAVs, researchers have improved several remarkable frameworks. Sommer et al. [6] promoted Faster R-CNN by training on self-acquired drone images and using median background subtraction before a region proposal network as extractors of regions of interest (ROIs). They successfully identified drones from birds and achieved the best performance on the drone-vs-bird challenge dataset on the 16th International Conference on Advanced Video and Signal Based Surveillance (AVSS 2017) then. Saqib et al. [23] conducted experiments with different CNN-based architectures including Zeiler and Fergus (ZF), Visual Geometry Group (VGG16), etc. The results showed that the VGG16 backbone performs better than others. Aker et al. [24] also used an end-to-end CNN architecture and created an artificial video dataset by combining real drone and bird images with different backgrounds. Carrio et al. [25] detected UAVs using depth maps, which are useful for collision avoidance, and generated a synthetic depth dataset by utilizing UAV AirSim software. Craye and Ardjoune [26] identified the ROIs using a U-Net network and used ResNetV2 for object classification. The proposed spatiotemporal method reports the best score on the AVSS2019 drone-vs-bird challenge. Magoulianitis et al. [27] proposed a method by jointly using super-resolution with object detection to address low resolution and showed that the detector can be maximally benefited from super-resolution when the two models are trained jointly. Next year, in the 3rd drone-vs-bird challenge on AVSS2020 [5], the Gradiant Team with refined Cascade R-CNN achieves the best performance. Koksal et al. [7] improved the training process and implementation of YOLOv3 by considering annotation errors and achieved a performance enhancement in detecting drones. Isaac-Medina et al. [9] first provided a benchmark for UAV detection and tracking for further research and compared the leading object detection frameworks.

In addition, there are effective solutions for other drone-like objects based on appearance, which are suitable for UAVs to some extent, such as the Glance-and-Stare Detection (GSD) method for birds [21], the YOLOv5 with the attention mechanism [28] for drone-captured objects, pyramid feature based framework [29] and perceptual Generative Adversarial Network (GAN) [30] for small objects. However, all the appearance-based methods above perform poorly under complex backgrounds or appearance similarity.

## 2.2. Motion-Based Methods

Motion-based methods, which rely on modeling inter-frame relations, are appearance invariant. Optical flow and motion modelling of foreground are the two main classes. Although motion features have been widely used for tracking [31], motion extraction [11], appearance feature propagation [15] and the Tracking-Before-Detection (TBD) schemes [32], there are still very few works that have been done for directly recognizing objects by their motion features.

"Deep" optical flow methods lead in performance of motion feature extraction. The Flownets [11,12] are the pioneer works. They used deep neural network to propagate appearance features across frames, so as to avoid costly feature extraction for non-key frames. On the basis of Flownets, many successive improvements [13] have been made to achieve high-accuracy optical flow estimation for both 2D and 3D cases. Zhu et al. [14] first proposed a deep optical flow-based end-to-end detection framework. The research [15] also presented a flow-guided feature aggregation framework. Bao et al. [16] introduced the Kalman filtering system to deep optical flow estimation, which improved the robustness and efficiency.

For detecting UAVs, Srigrarom et al. [18] identified drones from birds by in-plane trajectories with a Support Vector Machine (SVM) classifier and achieved over 80% accuracy on a self-collected video clip set. Some concepts in this work, such as the turning angle and the object pace, inspire the present research. Rozantsev et al. [19] accurately recovered 3D UAV trajectories using multiple fixed cameras, which can be used for tracking or detection. Most motion-based methods are used for region proposal or appearance feature propagation. To our knowledge, little work has been done on detecting drones by directly using intrinsic motion features, which is the start point of the proposed MUSAK method.

## 2.3. Hybrid Methods

Hybrid methods combined both motion and appearance features, thus usually get higher accuracy.

Schumann et al. [8] proposed a two-stage system. The first stage generated hypotheses by evaluating whether the current trajectory belongs to a UAV or to a random distractor. The following stage used a SSD network to perform further classification. Rozantsev et al. [20] introduced an object-centric motion compensation technique before a CNN-based network. The method eliminated the redundant region proposals by learning the motion features, and achieves significant improvements on two datasets for UAV and aircraft detection. Rodriguez-Ramos et al. [22] proposed an inattentional framework to extract a synthetic feature map containing the context features, which is further refined with Convolutional Long-Short Term Memory (ConvLSTM) layers. The method had a low time complexity and comparable detection accuracy.

In addition, there are other approaches for small object detection. Zhang et al. [33] used Faster R-CNN and a deep flow network named IRR-PWC to enhance the features and significantly reduced the false and missed objects. Liu et al. [34] used a Bi-directional Convolutional Long-Short Term Memory (Bi-Conv-LSTM) to learn the motion model and a 3D-Conv to solve the low contrast issue. Its experiments show that the trajectory accuracy is more than 90% under low signal-to-noise rate (SNR) scenarios, and it also keeps a low false alarm rate for infrared targets.

In general, the above methods take the motion features as the relation between appearance features or the appearance feature propagator. However, our method, MUSAK, is different to them, its key idea is to directly classify objects with kinematic parameters during objects' motion, although it also employs the LSTM network [22,34] as a classifier.

## 3. Multi-Scale Space Kinematic Method for Drone Detection

The proposed MUSAK method will be described in this section. Figure 1 gives the detection framework. The key idea of our method is to employ three groups of kinematic

parameters from 2D space, pseudo 3D space and 3D space, respectively, to deal with scenarios with different image qualities for detection.

Specifically, for each time-series input of ROIs, the first step of the MUSAK method is to employ the ground plane alignment method to calibrate the orientation. Then, according to the number and quality of keypoints in each ROI, MUSAK introduces three individual scale spaces to describe the target: the detail scale, the edge scale, and the block scale, exactly corresponding to the mentioned 3D space, the pseudo 3D space and the 2D space.

At the detail scale, the number of robust keypoints is enough to produce a high-precision 3D structure of the target, where the depth map and all the 3D kinematic parameters can be obtained. While at the edge scale, the robust keypoints are not enough to get a reliable depth structure, so we try to get a relative depth estimation from two neighboring frames, which is described by a set of relative 3D kinematic parameters, namely the pseudo 3D parameters, following the intuition "big when near". At the block scale, there are so few robust key points that can be found that the object depth cannot be calculated. Hence, the object motion is treated as movement on the pixel plane. Under such a condition, only the 2D kinematic parameters can be acquired. Loosely speaking, the method of three-scale division reflects the effect of different distance for the target under same optical observing system.

The following subsection will discuss more about each step of our method. Table 1 shows the notations used in this paper.

**Table 1.** Notations and definitions.

| Notation | Definition | Notation | Definition |
|---|---|---|---|
| $Q_{key}$ | Keypoints quality | $\mathbf{T}_t$ | Translation vector with time |
| $\boldsymbol{\rho}$ | Kinematic parameter combination | $\mathbf{R}_t$ | Rotation matrix with time |
| $\vec{v}$ | Translation velocity | $f$ | Focal length |
| $\vec{a}$ | Translation acceleration | $F$ | Frame rate |
| $\vec{\omega}$ | Angular velocity | $N$ | The feature number in a ROI |
| $\vec{\alpha}$ | Angular acceleration | $\sum_{pixel}$ | Total number of pixels in certain ROI |
| $s(t)$ | Scale measurement with time for an object in pseudo 3D space | $P, \mathbf{P}$ | Coordinates of point p in pixel frame and world frame |

### 3.1. ROI Extraction and Tracking

The ROI extraction is the first step. For its high robustness and adaptivity in tackling different backgrounds, Visual Background Extractor (ViBe+) [35] is employed for retrieving ROIs. The main steps are briefly as follows:

1.　Background model initialization.

For each pixel, a sample set is created, which consists of the pixel and its 20 neighbor pixels. Through the first frame, each pixel sample set is randomly selected from the adjacent 24 pixels.

2.　Background/foreground detection.

In ViBe+ method, each pixel compares to its pixel sample set to determine whether it belongs to background, if the cardinality of the set intersection of a given sphere of radius 30 and the sample set is above the given threshold of 2.

3.　Background model updating.

The updating process follows conservative policy, lifespan policy and spatial consistency. Each background point will be transferred to a foreground point with a probability of 1/rate, in which rate denotes the updating factor. As the background varies rapidly in our scenes, the rate here is set to 8 (a background pixel has one chance in 0.125 of being selected to update its model).

During implementation, the following modifications suggested in work [35] are also adopted: a different distance function and threshold criterion, a separation between up-

dating mask, and the detection of blinking pixels. The result of a typical motion ROI is illustrated in Figure 2.

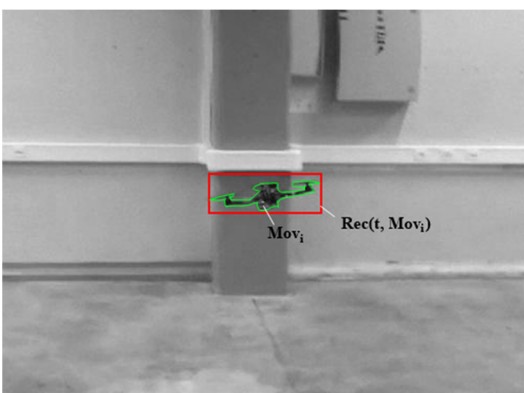

**Figure 2.** An illustration of a ROI.

As Figure 2 shows, $Mov_i$ indicates the $i$-th extracted motion ROI, and Rec ($t$, $Mov_i$) represents bounding box containing $Mov_i$ at time $t$. For each newly extracted motion ROI, Spatially Regularized Correlation Filter (SRDCF) [36] is adopted for inter-frame tracking due to its ability to address the tracking task efficiently and robustly.

In addition, this method aligns the direction of the ground plane by using the method [37] to calibrate the direction of the following kinematic parameters at the start of acquisition.

### 3.2. The Keypoints Criterion

The keypoints criterion physically divides the whole detection scales into three scale spaces, the detail scale, the edge scale and the block scale, based on the image quality or definition.

The examples of each scale space are shown in Figure 3. When the object is near, its detail is clear, and therefore plenty of robust feature points can be utilized. When it goes further, the details of the object vanish first, only the edges and the large part remain visible. When the distance becomes far enough, all robust feature points aggregate to a blurred pixel block. This physical transition thus yields three corresponding scales: detail, edge, and block scale. In the detail scale, the object possesses four or more robust feature points for interframe alignment. At the edge scale, the keypoints lose robustness and transfer to edge or corner points, which can partially retain structural completeness. At the block scale, as Figure 3c shows, the object becomes to a moving or still pixel block on the background.

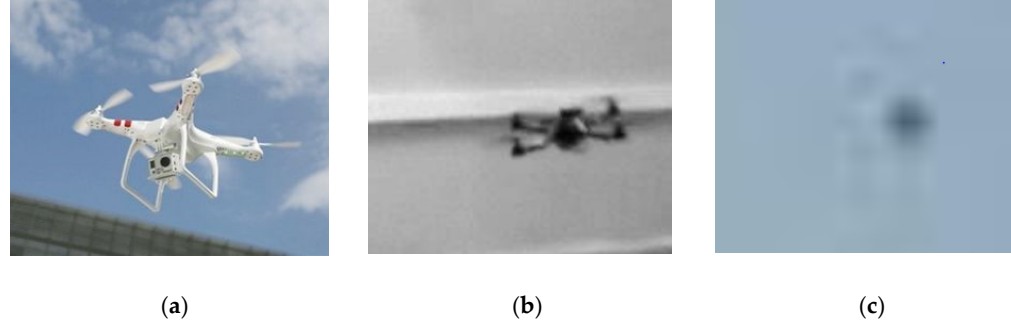

(**a**)                    (**b**)                    (**c**)

**Figure 3.** Example for each scale. (**a**) Detail scale. (**b**) Edge scale. (**c**) Block scale.

As stated above, keypoints quality measuring is the first and also the important step in our method. The quality of keypoints, denoted by $Q_{key}$, is modeled as follows:

$$Q_{key} = \frac{1}{\alpha \|N_{rob}\| + \|N_{edge}\|} \tag{1}$$

$Q_{key}$ ranges from 0 to 1, and lower value means higher keypoints quality. The measure describes the number and quality of keypoints. $N_{rob}$ denotes the number of robust feature points from the invariant feature descriptor (SURF), $N_{edge}$ is the number of corner and edge points and $\alpha$ is the relation coefficient describing how many robust feature points increase as the corner and edge points increase for an object. The value of $\alpha$ relies on the type of object and the detection condition. For the related objects (drones, cars, pedestrians and birds) in our detection scenarios, $\alpha$ is set to 25. Then, the keypoints criterion is written as

$$\begin{cases} Q_{key} \in \left(0, Q_{edge}\right), & \text{detail scale} \\ Q_{key} \in \left[Q_{edge}, Q_{block}\right), & \text{edge scale} \\ Q_{key} \in \left[Q_{block}, 1\right), & \text{block scale} \end{cases} , \tag{2}$$

where $Q_{edge}$ and $Q_{block}$ are threshold values for the edge and block scales. Based on our detection task, the values are set to $Q_{edge} = 1/500$ and $Q_{block} = 1/50$.

### 3.3. Extraction of 3D Kinematic Parameters

This section describes the extraction process of 3D kinematic parameters at the detail scale.

### 3.3.1. Depth Estimation

Depth value is the key to recovering the 3D structure of the ROI. Current methods for obtaining depth maps can be mainly classified into three kinds: laser measuring, stereo vision and image based estimation. Due to low cost and high accuracy, depth estimating from image is popular, and thus adopted in the present research. Specifically, we choose the ViP-DeepLab [38] method for its state-of-the-art performance among the current monocular depth estimation methods.

### 3.3.2. Parameter Extraction

Here, we will identify the 3D translation and rotation kinematic parameters, denoted as kinematic vector $\rho = (\vec{v}, \vec{a}, \vec{\omega}, \vec{\alpha})$ for $Mov_i$. $\rho$ is composed of translation velocity $\vec{v}$, acceleration $\vec{a}$, angular velocity $\vec{\omega}$ and angular acceleration $\vec{\alpha}$.

The reference frame here is a right-handed 3D camera coordinate system, which is also the world coordinate system, shown in Figure 4a, its origin is located at the optical center of the camera, the Z-axis points away along the optical axis in correspondence with the depth direction, the X-axis is consistent with the u-axis of the pixel coordinate system in Figure 4b, and the Y-axis is consistent with the v-axis of the pixel plane coordinate system.

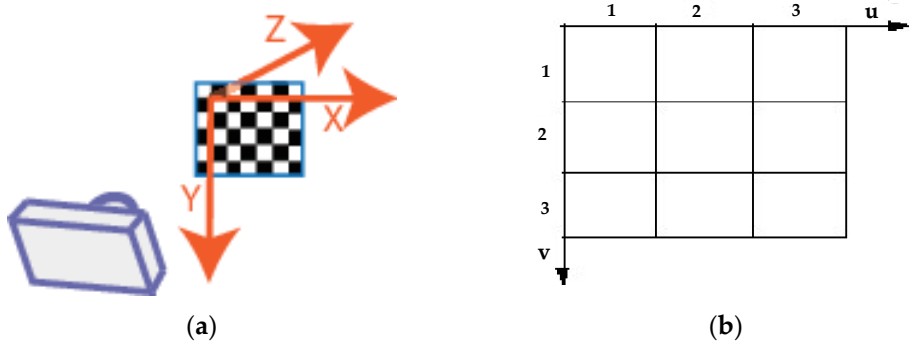

(**a**)　　　　　　　　　　　　　　　　　　　(**b**)

**Figure 4.** The reference frames. (**a**) 3D coordinate system. (**b**) pixel coordinate system.

Given a pair of pixel coordinates $P_t^i = \left(x_t^i, y_t^i\right)$ of the $i$-th feature point (SURF descriptor [39]) extracted in the area $Mov_m$ at current time $t$ and the corresponding depth value $d_t^i$, the aligned points in neighboring frame $t-1$ and $t+1$ are denoted by $P_{t-1}^i = \left(x_{t-1}^i, y_{t-1}^i\right)$ and $P_{t+1}^i = \left(x_{t+1}^i, y_{t+1}^i\right)$ with depth of $d_{t-1}^i$ and $d_{t+1}^i$. Moreover, the focal length is denoted by $f$, the frame rate by $F$ and the total feature number by $N$. Then, according to

the camera geometry, the coordinate of the *i*-th feature point in 3D Euclidean space is
$\mathbf{P}_i = \left( X_i^w, Y_i^w, Z_i^w \right) = \left( d_t^i x_t^i / f, d_t^i y_t^i / f, d_t^i \right)$.

According to the EPNP (efficient perspective-n-point) method [40], with more than four $(N \geq 4)$ aligned $(P^i, \mathbf{P}_i)$ pairs in $Mov_m$, we can determine the translation vector $\mathbf{T}_t = \left( T_t^x, T_t^y, T_t^z \right)'$ and rotation matrix $\mathbf{R}_t = \begin{bmatrix} R_{11} & R_{12} & R_{13} \\ R_{21} & R_{22} & R_{23} \\ R_{31} & R_{32} & R_{33} \end{bmatrix} = \begin{bmatrix} R'_1 \\ R'_2 \\ R'_3 \end{bmatrix}$ by solving the camera projection function

$$Z_j^w \begin{bmatrix} P^i \\ 1 \end{bmatrix} = \mathbf{K} \begin{bmatrix} \mathbf{R} & \mathbf{T} \\ 0 & 1 \end{bmatrix} \mathbf{P}_i = \begin{bmatrix} f_x & 0 & c_x \\ 0 & f_y & c_y \\ 0 & 0 & 1 \end{bmatrix} \begin{bmatrix} \mathbf{R} & \mathbf{T} \\ 0 & 1 \end{bmatrix} \sum_{j=1}^{4} \alpha_{ij} \begin{bmatrix} X_j^w \\ Y_j^w \\ Z_j^w \\ 1 \end{bmatrix}, \tag{3}$$

where $f_x, f_y, c_x, c_y$ are the intrinsic parameters of the camera and $\alpha_{ij}$ are homogeneous barycentric coordinates (Refer to [40] for more details).

Then, with the translation vector $\mathbf{T}_t$ and rotation matrix $\mathbf{R}_t$, the translation velocity $\vec{v}_t$ in terms of components is written as

$$\vec{v}_t = \left( \frac{dT_t^x}{dt}, \frac{dT_t^y}{dt}, \frac{dT_t^z}{dt} \right)^{appr.} = \left( FT_t^x, FT_t^y, FT_t^z \right). \tag{4}$$

By using the central difference, the acceleration $\vec{a}$ can be calculated as

$$\vec{a}_t = \left( F\left( v_{t+1}^x - v_{t-1}^x \right)/2, F\left( v_{t+1}^y - v_{t-1}^y \right)/2, F\left( v_{t+1}^z - v_{t-1}^z \right)/2 \right). \tag{5}$$

For the rotation matrix $\mathbf{R}_t$, the rotation angle $\varphi$ between two neighboring frames can be extracted by using Rodrigues transformation

$$\varphi = arc\cos\left( \frac{1}{2}[tr(\mathbf{R}_t) - 1] \right), \tag{6}$$

where $tr(\mathbf{R}_t)$ refers to the trace of $\mathbf{R}_t$. The anti-symmetric matrix $[\mathbf{u}]_x$ of rotation axis $\vec{u} = \left( u_x, u_y, u_z \right)$ (unit vector) is given by

$$[\mathbf{u}]_x = \begin{bmatrix} 0 & -u_z & u_y \\ u_z & 0 & -u_x \\ -u_y & u_x & 0 \end{bmatrix} = \frac{1}{2\sin\varphi} \left( \mathbf{R} - \mathbf{R}^T \right), \tag{7}$$

where $\mathbf{R}^T$ refers to the transposition of $\mathbf{R}$. Afterwards, the angular velocity $\vec{\omega}_t$ is obtained by

$$\vec{\omega}_t = F\varphi\left( u_x, u_y, u_z \right). \tag{8}$$

Additionally, by employing the central difference to angular velocity $\vec{\omega}_t$, we obtain angular acceleration

$$\vec{\alpha}_t = F^2(\varphi_{t+1} - \varphi_{t-1})\left( u_x, u_y, u_z \right)/2. \tag{9}$$

Thus far, all the 3D kinematic parameters at the detail scale have been obtained.

### 3.4. Extraction of 2D and Pseudo 3D Kinematic Parameters

As mentioned above, when there is a sufficient number of pixels in target ROIs, all of the 3D kinematic parameters can be retrieved. However, there are also many cases where the depth information cannot be obtained, or the robust aligned keypoints pairs $(N < 4)$

are not enough due to lacking pixels. So, inferring motion structure for this kind of objects degenerates to a 2D case.

This subsection extracts kinematic parameters at the edge and block scales. The reference frame is consistent with the pixel coordinate system in Figure 4b.

When the 3D motion structure degenerates to in-plane motion the feature point pairs $(P^i, \mathbf{P}_i)$ transfer to unaligned corner points or pixel blocks.

The 2D parameters at the block scale can thus be written as two derivative operators, acting on the in-plane translation vector $(T_t{}^u, T_t{}^v)$,

$$
\begin{cases}
\vec{v}_t\big|_{(u,v)} = \left( \frac{\partial}{\partial t} T_t{}^u, \frac{\partial}{\partial t} T_t{}^v \right) \\
\vec{a}_t\big|_{(u,v)} = \left( \frac{\partial^2}{\partial t^2} T_t{}^u, \frac{\partial^2}{\partial t^2} T_t{}^v \right)
\end{cases}.
\tag{10}
$$

We also employ the central difference for calculating the parameters during implementation as the 3D process.

At the edge scale, the exact depth value cannot be calculated from Equation (3). However, following the intuition that "big when near", information about the target distance is still accessible. To this end, the present work introduces a new parameter named relative depth, instead of the physical depth along the direction of optical axis, to describe the inference value of the target distance. The relative depth is defined by the total pixel number in the ROI:

$$
s(t) = \beta_e \sum_{pixel}, \ \forall pixel \in Mov_i,
\tag{11}
$$

where $s(t)$ is also a measure of scale, $\sum_{pixel}$ is the total number of pixels in $Mov_i$ and $\beta_s$ is the compatibility parameter for connecting the kinematic parameters of other spaces. $s(t)$ models the object scale and is an approximation of the 3D depth. We refer to the newly reconstructed 3D structure as pseudo 3D space. Then, the pseudo 3D space is denoted as $\tau \cdot (u, v) \otimes \mathbf{S}$, where $\otimes$ denotes the Cartesian product, $\mathbf{S}$ is the reconstructed scale dimension and $\tau$ is the stretch factor introduced for dimensional balance, given by $\tau = s(t)/f$.

Then, the translation vector is modeled as

$$
\vec{v}(m)\big|_{\tau \cdot (u,v) \otimes \mathbf{S}} = \left( \tau \frac{\partial}{\partial t} T_t{}^u, \tau \frac{\partial}{\partial t} T_t{}^v, \frac{ds(t)}{dt} \right).
\tag{12}
$$

Similar to Equations (4) and (5), we can calculate the translation parameters for the edge scale. The "pseudo" here means an approximation caused by scale transition.

At the edge scale, rotation parameters are obtained from an estimation based on edge alignment. The robust feature point alignment degenerates to edge alignment due to scale transition, as shown in Figure 3. According to [41,42], $e_1$ and $e'_1$ (aligned edge pairs from neighboring frames, as Figure 5 shows) can match each other by the epipolar constraint and region-growing strategy. A high-accuracy matchup will induce decent aligned edge point pairs, which results in a convergent optimization for Equation (3).

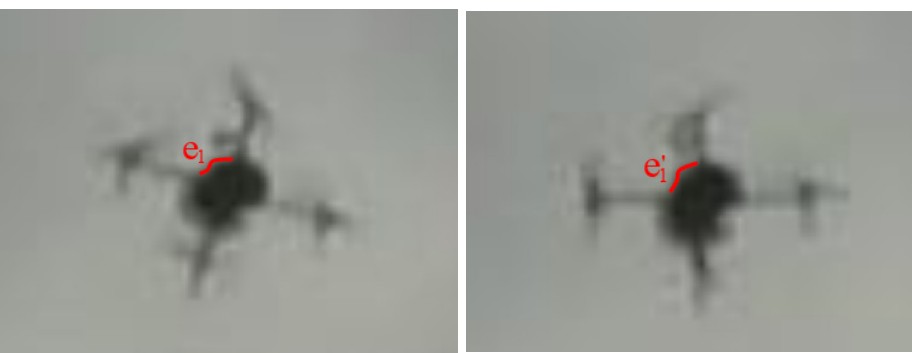

**Figure 5.** Illustration of edge alignment ($e_1$ and $e'_1$ are aligned edges from neighboring frames).

At last, the rotation parameters under the block scale cannot be calculated because almost all key points or edges contract into a blurred pixel block. Under such conditions, the rotational patterns vanish, and only the translation parameters are involved for recognition.

### 3.5. The Compatibility for Three Scale Spaces

The above sections have achieved kinematic parameters extraction for the introduced three scale spaces. This subsection will demonstrate the compatibility and relationship between the parameters.

The corresponding three extraction processes are uniformly illustrated in Figure 6. $O_1, O_2, O_3$ are three neighborhoods from the motion of a certain drone under the detail, edge and block scales, respectively. The kinematic parameters extracted in $O_1, O_2, O_3$ are based on the local 3D, pseudo 3D and 2D coordinate systems. $\phi_i$ is the map from the input motion neighborhoods to the local space. $c(t)$ is the time-parameterized motion process.

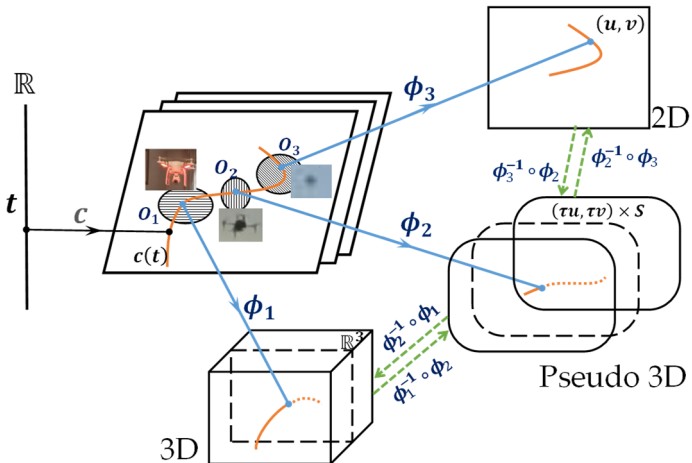

**Figure 6.** Illustration of the compatibility.

From the perspective of geometry, we can find three pairs of homeomorphic maps $\phi_i, i = 1 \ldots 3$ and the corresponding coordinate patch $O_\alpha \subset M, \alpha = 1 \ldots 3$, namely, a chart $(O_i, \phi_i)$, for the three scales, which gives birth to three coordinate patches, the local 3D Euclidean space, the local pseudo 3D space and the local 2D Euclidean space. Then, the map $\phi_i$ is

$$\begin{cases} \phi_1 : O_1 \mapsto (X, Y, Z) \in \mathbb{R}^3 \\ \phi_2 : O_2 \mapsto (\tau u, \tau v, s) \in \mathbb{R}^3 \\ \phi_3 : O_3 \mapsto (u, v) \in \mathbb{R}^2 \end{cases}. \tag{13}$$

When a drone belongs to the intersection of two or three coordinate patches, which means that the motion is on the boundary of related scale ranges, the extraction results under corresponding scales must meet the compatibility requirements. Then, the kinematic parameters at intersection neighborhoods should satisfy the following two conditions:

$$\begin{cases} \forall p \in O_1 \cap O_2, \ \phi_1(p) = \phi_2(p) \\ \forall p \in O_2 \cap O_3, \ \phi_2(p) = \phi_3(p) \end{cases}. \tag{14}$$

The first condition is

$$\forall p \in O_1 \cap O_2, \ \phi_1(p) = \phi_2(p). \tag{15}$$

Substituting the translation parameters for the map, that is

$$\forall p \in O_1 \cap O_2, \ (\tau T_t{}^u, \tau T_t{}^v, s(t)) = \left( T_t^x, T_t^y, T_t^z \right). \tag{16}$$

Equation (16) can be written in componentwise as

$$
\begin{cases}
\tau T_t{}^u = T_t^x \\
\tau T_t{}^v = T_t^y \\
s(t) = T_t^z
\end{cases}
\Rightarrow s(t)\big|_p = T_t^z.
\tag{17}
$$

Then, we get

$$
\beta_e = \frac{T_t^z}{\sum_{pixel}}.
\tag{18}
$$

Following the same fashion as the first condition, we can draw from the second one

$$
\forall p \in O_2 \cap O_3, \ \phi_2(p) = \phi_3(p).
\tag{19}
$$

i.e. $(\tau T_t{}^u, \tau T_t{}^v, s(t)) = (T_t{}^u, T_t{}^v, C)$

$$
\Rightarrow
\begin{cases}
\tau T_t{}^u = T_t{}^u \\
\tau T_t{}^v = T_t{}^v \\
s(t) = C
\end{cases}
\Rightarrow s(t)\big|_p = f
\tag{20}
$$
$$
\Rightarrow \beta_e = \frac{C}{\sum_{pixel}}.
$$

So far, for any motion $p$ belonging to the intersection $O_1 \cap O_2$, $s(t)$ equals the value of translation $T_t^z$ at $p$ and for the points in the intersection $O_2 \cap O_3$, $s(t)$ equals the focal length $f$ (under the homogeneous coordinates). The two requirements ensure the compatibility of the kinematic parameters extracted from three scales. The requirements also indicate that from the detail stage to the block stage, the structural degeneration process occurs, which causes the disappearance of feature points.

### 3.6. Drone Detection by GRU Network

Following the above process, the target has been abstracted as time-series kinematic parameter sets $(\vec{v}_t, \vec{a}_t, \vec{\omega}_t, \vec{\alpha}_t)$. After that, GRU networks [43] are employed to solve the following drone recognition problem.

More specifically, as shown in Figure 1, three GRU networks are employed for three scales. Each network is trained separately and is comprised of the update gate and reset gate. The three input vectors are in the same form of $\mathbf{x} = \left( \vec{v}_t, \vec{a}_t, \vec{\omega}_t, \vec{\alpha}_t \right)$ but differ in terms of components. The output is encoded by the one-hot vector, and the object classes include "drones", "birds", "pedestrians", "cars" and "others", which are the common moving objects presented in UAV scenes. In addition, the loss function is the cross entropy

$$
L = -\sum_i y_i \log \hat{y}_i,
\tag{21}
$$

where $y_i$ is the ground truth value and $\hat{y}_i$ is the prediction.

The GRU unit we adopt has the same structure as the one in the literature [43], containing the reset and update gates. The current activation is controlled by delivering the previous activation based on the calculation of reset and update gate.

### 3.7. The Hybrid MUSAK

Besides the motion feature, appearance-based DNN methods are with no doubt the most commonly used in object detection tasks, if the target is clearly visible. By decision fusion, the proposed MUSAK method is able to cooperate with appearance-based methods, such as Faster R-CNN, to enhance its performance. A simple but effective way to implement this idea is to output the final detection probability by averaging the confidence scores from MUSAK and Faster R-CNN.

By the fashion of average fusion, this paper employs the appearance-based method Faster R-CNN [23] to form the hybrid MUSAK method. We choose the VGG16 architecture as the backbone and fine-tune the network on our MUD dataset. To improve recall per-

formance, we set 15 types of anchors with five scales (2,4,8,16,32) and four ratios (0.5,1,2). More performance analysis about the hybrid MUSAK will be discussed in Section 4.

## 4. Experiments

In this section, the performance of MUSAK method is evaluated and compared with other existing methods. Further analysis on MUSAK is also conducted.

### 4.1. Setup and Datasets

The existing datasets for drone detection pay more attention to the diversity of appearance and illumination, and to some degree, they are too clean, which means other important issues, such as multiple drone-like distractors, scale transitions and motion complexity, have not been considered.

To address the issue, we first construct a new dataset, named Multiscale UAV Dataset (MUD). It is comprised of the popular dataset Drone-vs-Bird [4,5], MAV-VID [22], Anti-UAV [44], part of the UAV-Aircraft Dataset (UAD) [20], and several self-made video clips having complex backgrounds or multiple drone-like objects as interference. MUD is designed to perform better training/evaluation for drone detection methods working in the real environment.

Specifically, MUD adds drone flight videos of indoor scenes, urban scenes and wild scenes. The newly added data not only contain basic annotations such as target categories and rectangular boxes but also mark the depth information, flying height, camera angle, and motion parameters for motion-based method research. (The dataset will be available soon on [45]). Example pictures in MUD are shown in Figure 7. The main acquisition equipment includes cameras (FE 24–240 mm), GPS (with precision of ±1.0 m) and laser range finder (0.2–50 m with ±0.5 cm).

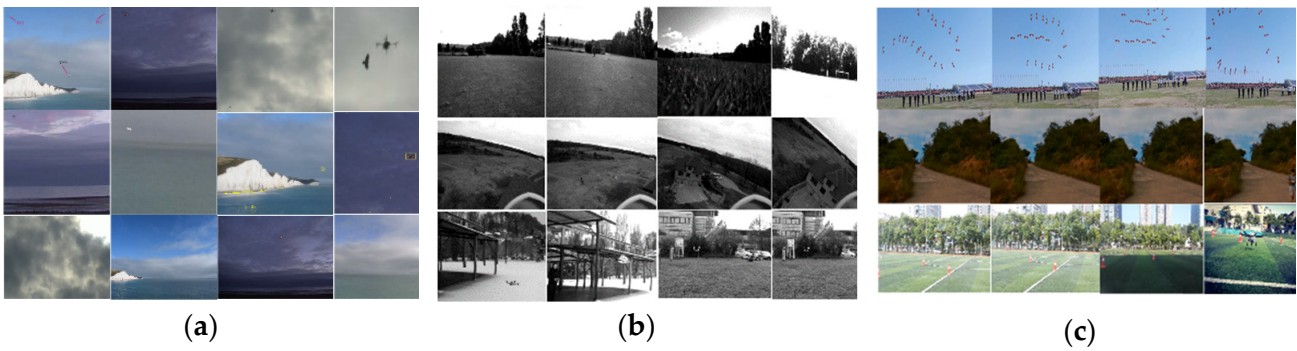

(a)        (b)        (c)

**Figure 7.** Example images in MUD dataset. (**a**) Drone-vs-Bird Dataset. (**b**) UAV-Aircraft Dataset. (**c**) Self-collected data.

In addition, we introduce simulation data from AirSim software for training MUSAK, which has also been used in training appearance-based methods [25].

### 4.2. Extraction Results of Motion ROIs

We compare different motion extraction methods based on 10 randomly selected groups of video clips with the ground truth of 250 ROIs. Using the number of bounding boxes (BB), recall and precision rate as indicators, the result is shown in Table 2.

**Table 2.** Comparison of different ROI extraction methods.

| Methods | AVG # of BBs | AVG Recall | AVG Precision |
|---|---|---|---|
| Frame Difference | 521 | 0.832 | 0.482 |
| Gaussian Mixture | 380 | 0.784 | 0.650 |
| Optical Flow | 370 | 0.853 | 0.620 |
| ViBe+ | 301 | 0.868 | 0.830 |

ViBe+ achieves the best performance in both recall and precision metrics. The implementation parameters are as follows: five pixels for minimum size of the foreground hole, 10 for sample size per pixel, eight for updating factor, while other parameters remain default as them in the previous work [35]. The exemplar results of motion ROI extraction via ViBe+ are illustrated in Figure 8.

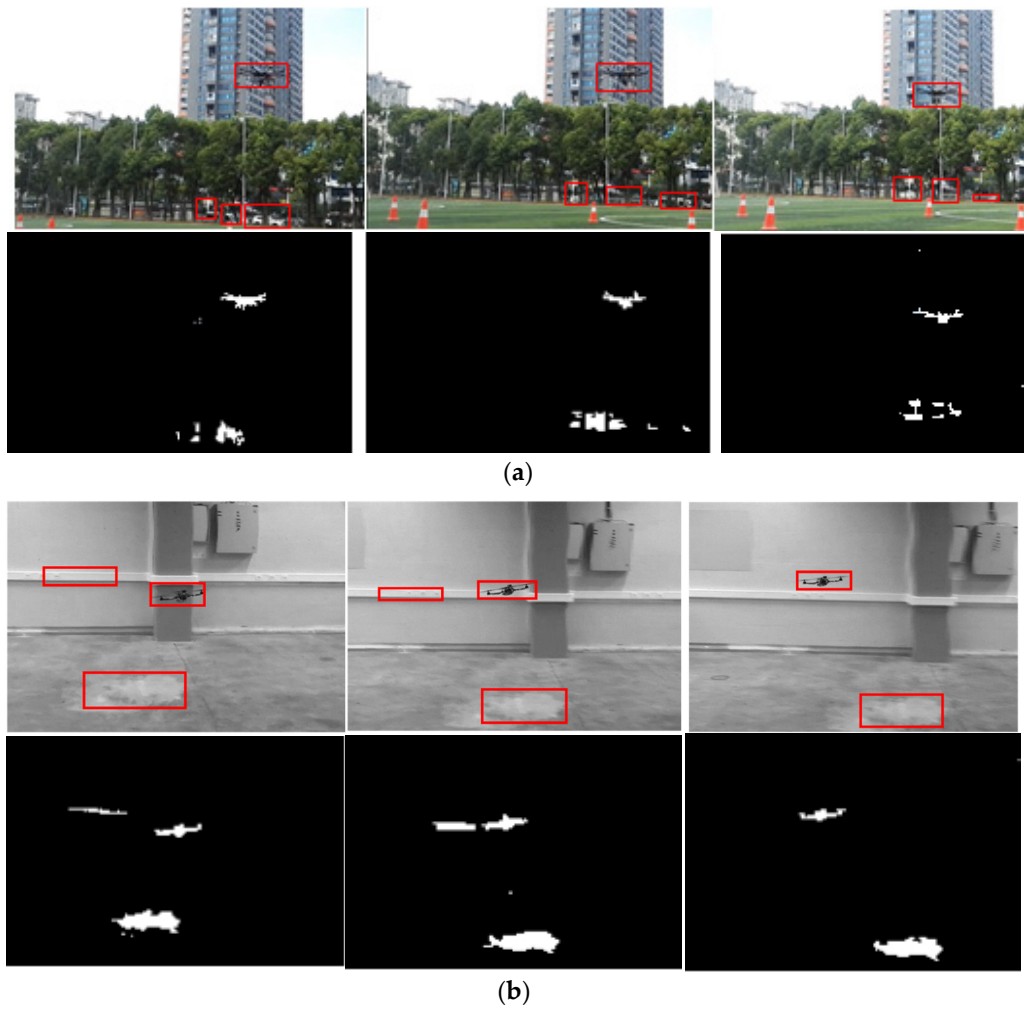

(a)

(b)

**Figure 8.** Illustrations of ROIs extraction. (**a**) Urban scene. (**b**) Indoor scene.

Figure 8 visualizes the extraction results of ROIs in an urban and indoor scene. The red windows in the top rows are the ROI bounding boxes, and the mask areas in the bottom rows are the ROIs.

Other motion extraction methods can also be employed as an alternative based on the target attributes.

### 4.3. The Extracted Kinematic Parameters

As described in Section 3.3, the kinematic parameters $\rho = \left( \vec{v}, \vec{a}, \vec{\omega}, \vec{\alpha} \right)$ in 3D space can be extracted. Compared with the ground-truth value given by the UAV's real-time kinematic differential system (RTK) and motion capture system (MCS), the extraction biases are shown in Figure 9. The onboard GPS ensures a vertical positioning accuracy of $\pm 0.5$ m and a horizontal positioning accuracy of $\pm 1.5$ m. Moreover, if the visual module on board also works, the vertical positioning accuracy is $\pm 0.3$ m, while the horizontal positioning accuracy is $\pm 0.5$ m.

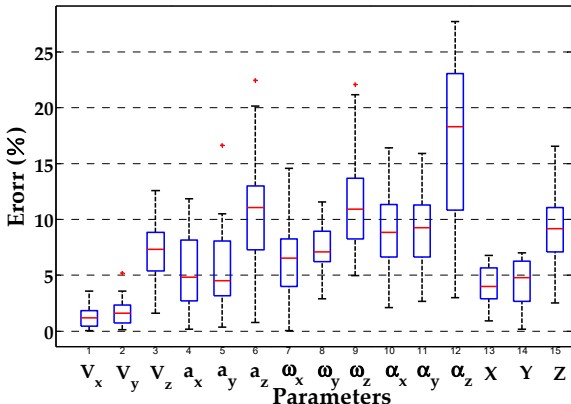

**Figure 9.** Biases of the extracted kinematic parameters.

Figure 9 shows the estimation biases for the kinematic parameters in terms of components. The average biases for $v, a, \omega, \alpha$ are 4.5%, 7.7%, 9.1% and 15.4%. Overall, the estimation of velocity $\vec{v}$ has the highest accuracy, while the angular acceleration $\alpha$ has the lowest accuracy. The translation biases are lower than the rotation biases, and the biases of the first-order parameters are lower than those of the second-order parameters.

The estimation of rotation parameters is more sensitive to feature alignment. A minor feature point shift on the pixel plane may cause large biases in the extracted rotation parameters. Therefore, the rotation parameters are relatively inaccurate compared with translation parameters. In addition, the bias of the second-order parameters is larger than the first order ones, since they are derived from the first-order parameters and the bias is added up to the estimation.

### 4.4. Drone Detection Results with MUSAK Method

In this subsection, we evaluate our MUSAK method and compare the proposed methods with other existing methods.

#### 4.4.1. Detection Results by GRU

During the initialization, we set the input dim to $4 \times Dim$, where 4 refers to the four parameters $v, a, \omega, \alpha$ and $Dim = 3$ refers to the dimension of the space. The batch size was 128, and the hidden layer number was 64. In the training process, we re-estimate the accuracy indicator for each epoch and fine-tune the hyperparameters, including the time step and bias. A low loss value should be ensured for each training batch. To eliminate the cumulative error of the system, the whole process will be initialized every 60 s. Figure 10 shows the examples of the detection results in three scales and a scene with tiny objects. The outputs include bounding boxes and scores.

Figure 10a–c shows the detection results in three different scenes, the indoor, fully air, and urban scene, which are under detail scale, block scale and edge scale, respectively. Our method is devoted to tackling multiclass classification, which includes five commonly presented classes of drones, birds, pedestrians, cars and others. Different classes of objects are marked in different colors with the class scores. The first row of images is feature-rich, while the last two rows have relatively poor appearance features. Though lacking appearance features, MUSAK can still recognize moving drones during flight.

Figure 10d presents the detection result for tiny objects (area < 100 px) with three kites as distractors, which is a Gordian knot to the previous methods. When a typical motion pattern appears, the confidence score of being a certain class will increase significantly, and the score for other classes will gradually decrease. The related box is considered to contain the object if the score exceeds 0.5. Table 3 presents the confusion matrix of the corresponding classes.

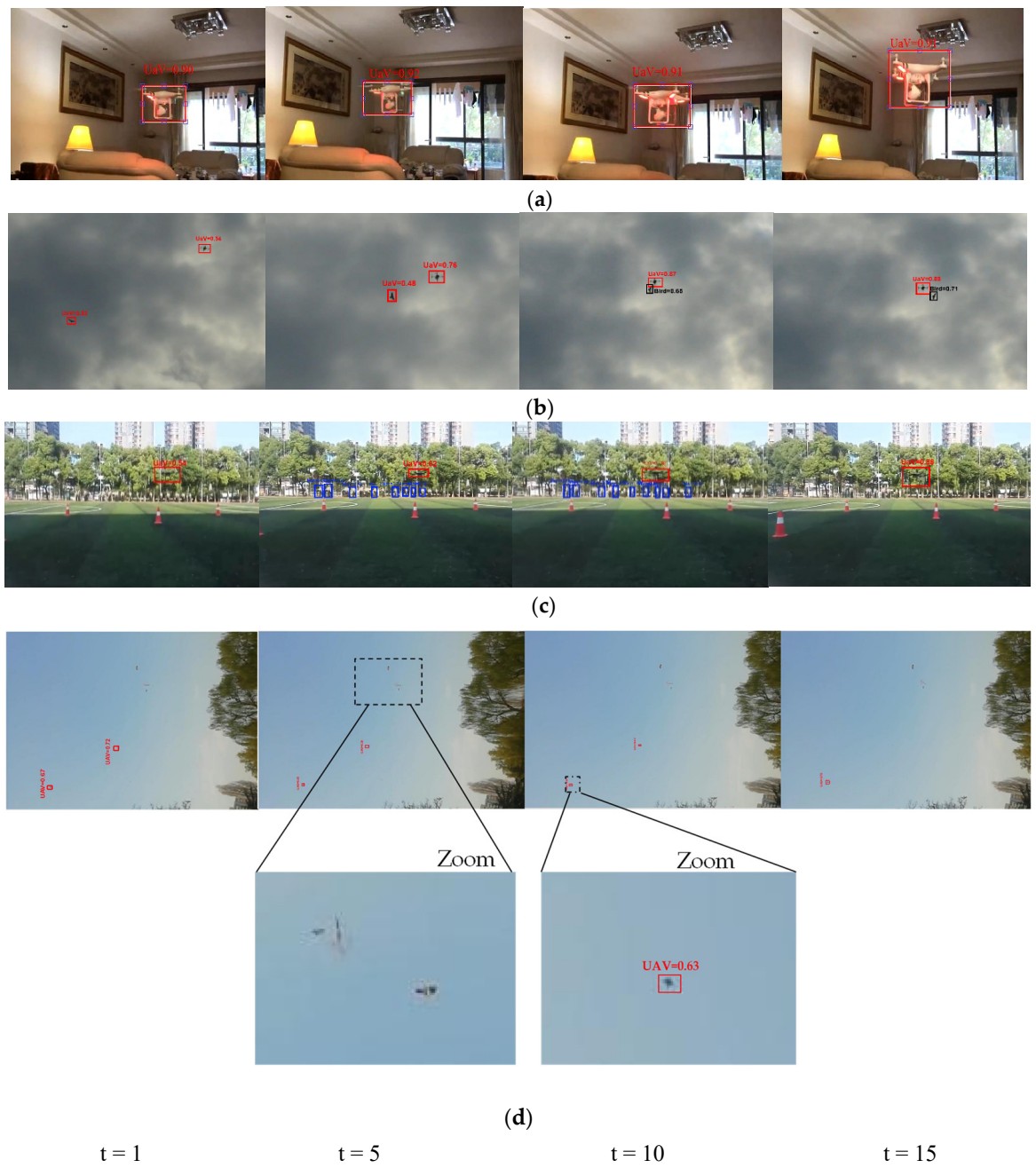

t = 1          t = 5          t = 10          t = 15

**Figure 10.** Detection results by MASAK method. (**a**) 3D. (**b**) 2D. (**c**) Pseudo 3D. (**d**) Tiny objects with drone-like disturbances.

**Table 3.** Confusion matrix of MUSAK by GRU.

| Predicted \ True Label | Drones | Birds | Pedestrians | Cars | Others |
|---|---|---|---|---|---|
| Drones | 0.68 | 0.25 | 0.02 | 0.01 | 0.11 |
| Birds | 0.19 | 0.58 | 0.01 | 0.00 | 0.10 |
| Pedestrians | 0.01 | 0.01 | 0.75 | 0.06 | 0.09 |
| Cars | 0.01 | 0.00 | 0.10 | 0.80 | 0.08 |
| Others | 0.11 | 0.16 | 0.12 | 0.13 | 0.61 |

The confusion matrix of the five classes (drones, birds, pedestrians, cars, and others) is shown in Table 3. The numbers in the table represent the rate of the number of true positive

samples to the total number. It can be found from the table that the detection accuracy of drones, pedestrians and cars is high, while the accuracy of birds is the lowest with a high confusion rate. Compared with birds, drones have a higher detection accuracy. Accuracy of pedestrians and cars is the highest. This is mainly because of their low motion complexity and high motion stability. In general, all the true positive rates (the diagonal elements) in the table can reach more than 58%.

### 4.4.2. Comparison with the Existing Methods

To compare the performance of different methods, referring to the literature [46], we take the AP values (AP, AP$_S$, AP$_M$ and AP$_L$) and PR (Precision-Recall) curve as metrics. Two specifically proposed metrics, the 95% point and tail gradient, are presented to describe the tendency of PR curves. The 95% point or knee point refers to the location where the PR curve falls to 95% of the highest precision value for a certain method, which indicates the precision and robustness of the method. The tail gradient is the absolute slope of the line linking the knee point and the point with precision of 0.1 on the curve, which also implies the robustness of methods.

We first present the PR curves of our methods and the major existing CV methods for drone detection, which are shown in Figure 11. The PR curves are under IoU = 50% (Intersection-over-Union). In addition to the motion-based method MUSAK and the hybrid MUSAK described in Section 3.7, other 7 controlled methods are involved. They are: FlowNet [13] based on deep optical flow features, Srigrarom et al. [18] using trajectory features, Faster R-CNN for drones [23], Craye&Ardjoune [26] using a combined framework with U-net and ResnetV2, Rodriguez-Ramoset al. [22] based on an inattentional ConvL-STM, Kim et al. [47] based on Yolov3 with attention mechanism and Rozantsev et al. [20] exploiting an improved Faster R-CNN with camera compensation. Generally, along with recall value increases, the PR curve for each method starts with a steady initial stage, in which the precision stays at a high level. When passing its knee point, it begins a fast falling process and ends at a low precision. Further comparison is listed in Table 4.

Each curve in Figure 11 can be divided into two parts: head and tail, divided by the 95% point (knee point). The head part lies on the left of the curve, comprised of the points with precision of over 95% of the highest precision of each curve. The appearance-based ([23,26,47]) and hybrid ([20,22] and Hybrid MUSAK) methods have a higher precision at the head part compared with motion-based methods ([13,18] and MUSAK). This is because the target in the head part is always feature-rich and easily detected by appearance. Since the Faster R-CNN based methods perform well on head part objects, such as Rozantsev [20] and Faster R-CNN [23], our hybrid MUSAK, which combines the MUSAK and Faster R-CNN, achieves the highest head precision with 0.92, a 2.2% increase compared with the previous SOTA method. For the motion-based methods, MUSAK (0.73) surpasses FlowNet [13] (0.69) and Srigrarom [18] (0.71) in its refined kinematics-based framework.

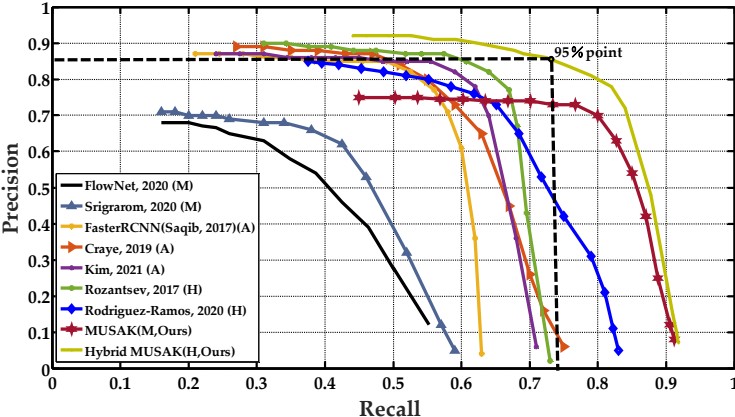

**Figure 11.** Methods comparison by PR curves (M: motion-based method; A: appearance-based method; H: hybrid method).

**Table 4.** Methods comparison by the metrics.

| Type | Methods | AP | 95% Point | Tail Grad | $AP_L$ | $AP_M$ | $AP_s$ |
|---|---|---|---|---|---|---|---|
| Motion -based | FlowNet [13] | 0.312 | 0.30 | **2.37** | 0.322 | 0.243 | 0.212 |
| | Srigrarom et al. [18] | **0.356** | **0.38** | 2.78 | 0.474 | 0.251 | 0.230 |
| | DelMarco&Webb [17] | 0.296 | 0.34 | 2.89 | 0.294 | 0.205 | 0.193 |
| | MUSAK (ours) | **0.656** | **0.78** | 5.54 | 0.752 | **0.584** | **0.421** |
| Appearance -based | Gökçe et al. [48](LBP) | 0.418 | 0.54 | **2.96** | 0.546 | 0.202 | 0.034 |
| | Craye&Ardjoune [26] | **0.633** | 0.52 | **6.68** | 0.792 | 0.605 | **0.164** |
| | Magoulianitis et al. [27] | 0.588 | 0.50 | 7.45 | 0.681 | 0.620 | 0.106 |
| | Koksal et al. [7] (YOLOv3) | 0.468 | 0.52 | 10.20 | **0.861** | 0.285 | 0.102 |
| | Faster R-CNN [23] | 0.486 | 0.57 | 11.07 | **0.841** | 0.185 | 0.134 |
| | Grandiant [5] (Cascade R-CNN) | 0.501 | 0.59 | 8.51 | **0.812** | 0.384 | 0.131 |
| | Tian et al. [21] (3D CNN) | 0.502 | 0.60 | 8.12 | 0.707 | 0.414 | 0.125 |
| | Zhu et al.[28] (YOLOv5) | 0.612 | **0.62** | **7.82** | 0.812 | 0.542 | **0.185** |
| | Kim et al. [47] (YOLOv5) | 0.566 | 0.62 | **6.45** | 0.832 | 0.512 | 0.164 |
| | Liu et al. [29] | 0.545 | 0.58 | 7.94 | 0.721 | 0.511 | **0.191** |
| Hybrid | Schumann et al. [8] | 0.616 | **0.71** | 10.60 | 0.794 | **0.650** | 0.232 |
| | Rozantsev et al. [20] | 0.631 | 0.65 | 13.10 | 0.813 | 0.372 | 0.312 |
| | Rodriguez-Ramos et al. [22] | **0.688** | 0.55 | **2.34** | **0.863** | 0.547 | **0.340** |
| | Zhang et al. [33] | 0.572 | 0.59 | 7.70 | 0.717 | 0.441 | 0.320 |
| | Hybrid MUSAK (ours) | 0.785 | 0.74 | 6.60 | 0.856 | **0.648** | **0.434** |

As for the tail part, we calculate the tail gradient (TAIL GRAD for short) to evaluate the degradation speed of the method. A large tail gradient indicates severe degeneration of performance. The results are listed in Table 4. It can be found that the appearance-based methods have larger tail gradients than motion-based methods, which reveals that motion features are more robust. The tail gradient for MUSAK (5.54) is relatively lower than that of appearance-based methods but higher than that of Srigrarom [18] (2.78). This is because Srigrarom [18] focuses on trajectory features, which belong to long-term motion characteristics, while our MUSAK method extracts both short-term and long-term motion features. The short-term features for 3D space depend on precise keypoints alignment, which is often hard to meet. Furthermore, the highest score on precision of Srigrarom [18] is lower.

The location of the 95% point is also an overall metric to expose the precision and robustness of a method. The top right location of the 95% point demonstrates a relatively high precision and strong robustness. The 95% points of previous motion-based methods are at recall < 0.40. The 95% points of previous appearance-based methods range from recall = 0.45 to recall = 0.63. All these methods are on the left of our MUSAK and hybrid MUSAK. MUSAK promotes the 95% point of motion-based methods from recall = 0.38 to recall = 0.78, which is a more than 2x enhancement. Hybrid MUSAK promotes the value by 15.4% from recall = 0.65 to recall = 0.75. The enhancements result from the well-directed motion-based detection scheme.

In addition to the metrics mentioned above, Table 4 also introduces more metrics named $AP_S$, $AP_M$ and $AP_L$, which refer to the AP for small, medium and large objects. $AP_L$ should be larger than AP while $AP_S$ is smaller than AP and larger AP values represent higher accuracy. In general, motion-based methods have higher $AP_S$ values than appearance-based methods, while for $AP_M$ and $AP_L$, the values of appearance-based methods are higher. Craye & Ardjoune [26] achieved the highest AP among appearance-based methods for its refined preprocess before classification. The proposed MUSAK presents the highest AP value of 0.656 for motion methods, which improves the previous methods by more than 84.3% due to its well-directed modeling of the motion process. The hybrid MUSAK reports state-of-the-art performance with 0.785 AP value, an at least 14.1% increase compared to the previous SOTA method (Rodriguez-Ramos [22]). Compared with other hybrid methods, promotion is mainly achieved from the improvement on detection of small and medium objects.

## 5. Further Analysis for MUSAK

To further reveal the adaptivity and significance of our methods, the keypoints quality, temporal-spatial resolution and significance of the kinematic parameters are taken into consideration in this section.

### 5.1. The Impact of Keypoints Quality

Keypoints quality, which is described by the $Q_{key}$ introduced in Section 3.2, demonstrates the definition of objects. The low value indicates a high keypoints quality as well as the object definition. Then, the impact of keypoints quality on the detection methods is presented as the correlation between AP and $Q_{key}$ plotted in Figure 12.

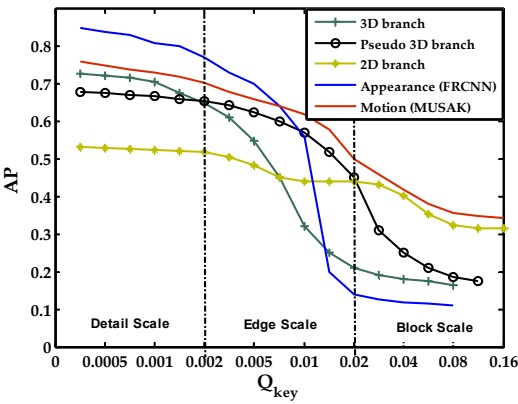

**Figure 12.** The impact of keypoints quality.

In Figure 12, four curves of AP with respect to $Q_{key}$ for the appearance-based method (Faster R-CNN) and MUSAK (three detection branches and the synthesized) illustrate how the object definition impacts the performance of the methods.

In general, all curves are in a falling process due to the degeneration of keypoints. The 3D motion, pseudo 3D motion and appearance curves are of a S-shaped falling process, a slight decline in the head part followed by a slump tail. The 2D motion curve is a wave-like falling process without any dramatic drops, which indicates that the 2D motion detection branch is more robust to keypoints quality. More specifically, the appearance curve has the highest AP value of 0.95 in the head part but the maximum downward gradient in the tail part, and ends at the lowest value (0.11). The 3D/pseudo 3D branch curve possesses a relatively high AP value, approximately 20% less than the head value of the appearance curve, but a relatively slow falling subprocess and a 90% increase for the ending value. The 2D motion curve starts at a lower AP value of 0.53, then with a middle stage AP of approximately 0.44 and ends at 0.315, a 3-fold increase in the appearance method value because the 2D motion detection branch can extract the appearance-invariant motion features. In summary, all methods are impacted by keypoints quality to some extent. The appearance method is more sensitive to keypoints quality, while MUSAK is more robust, which results in better performance in low-definition scenarios.

### 5.2. The Impact of Temporal-Spatial Resolution

For further analyzing the performance of MUSAK, we consider the temporal-spatial characteristic and focus on each detection branch. The impact of the temporal-spatial resolution on different detection methods and branches is shown in Figure 13, in which the time resolution is represented by the frame rate, while the spatial resolution uses the positioning accuracy (absolute relative error) as a measurement. Videos with different frame rates and positioning accuracy are collected for calculating the AP value.

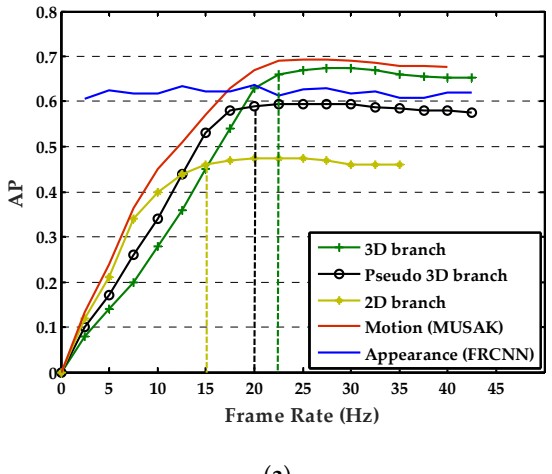

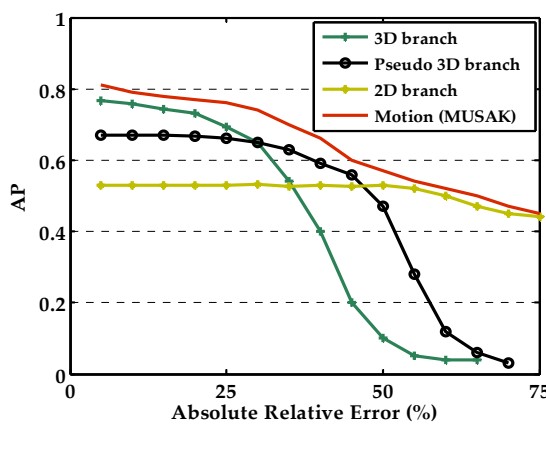

(**a**)          (**b**)

**Figure 13.** The impact of temporal-spatial resolution. (**a**) Impact of Frame Rate. (**b**) Impact of Absolute Relative Error.

Figure 13a presents the relationship between AP and frame rate (the motion blur is eliminated). In general, the curves of motion-based methods are composed of a rapid incline head process followed by a stable ending process with a slight decline tail due to the forgetting gates in GRU for handling long sequence, while the appearance curve is relatively stationary. The 2D motion curve has the maximum upward gradient and first reaches its peak at 0.47 at 15 fps. The pseudo 3D and 3D motion methods reach peak AP values of 0.59 and 0.67 at 20 and 23 fps, respectively. It can be concluded that the optimal frame rate is close to 25 fps.

Figure 13b presents the relationship between AP and positioning accuracy. The curve for the 3D branch starts to decline significantly when it goes over the point of error = 25%, while for pseudo 3D, this critical point is approximately error = 45%. The 3D detection branch is most sensitive to spatial resolution, while the 2D detection branch is more robust. The 2D and pseudo 3D detection branches compensate the loss caused by positioning error for MUSAK.

Based on the above results and analyses, it can be concluded that MUSAK deals with moving objects under different backgrounds and enhances the performance for dim drones. MUSAK requires object tracks and coordinates transformation from camera coordinate frame to world coordinate frame to retrieve the relative motion process for objects. For moving cameras which will enlarge the vision field, the motion compensation should be introduced.

### 5.3. Significance of the Kinematic Parameters

In this part, we conduct an ablation experiment to analyze the significance of the kinematic parameters. Without using all the parameters mentioned in Section 3.3, here, different combinations of kinematic parameters from single parameter to multi-combinations are adopted by MUSAK for detection. The results are presented in Figure 13.

Figure 14a shows the drone detection results with a single kinematic parameter, The X, Y and Z axes are the same as the definition in Section 3.3.2. It can be seen from Figure 14a that the AP values of the second-order (acceleration and angular acceleration) parameters are higher than those of the first-order (velocity and angular velocity) parameters, and it is also obvious that the AP of the Y components for translation parameters (velocity and acceleration) are the highest compared with the X and Z components, while the Z components are highest for rotation-related parameters (angular velocity and angular acceleration). Detection by the Y component of acceleration achieves the top AP followed by the Z component of angular acceleration. Referring to the coordinate declaration in Section 3.3.2, two important conclusions can be obtained. Firstly, the translation parameters

in the gravity direction and rotation parameters in the direction of the optic axis are of great significance. Secondly, the second-order kinematic parameters carry more motion characteristics for drones.

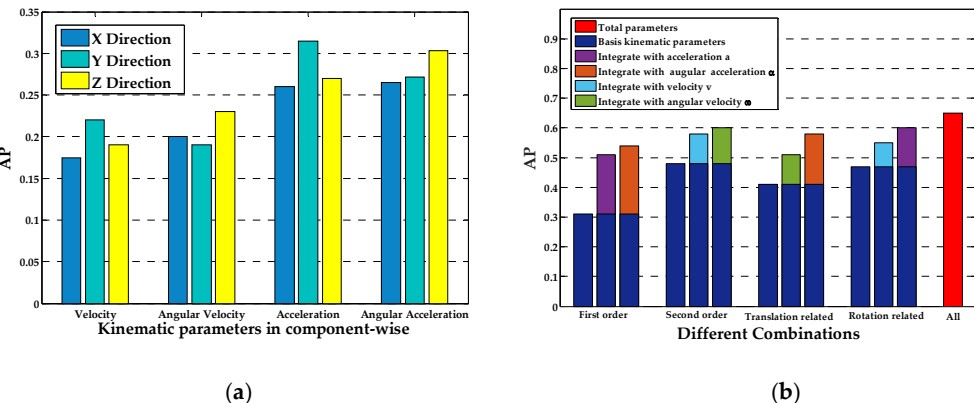

**Figure 14.** Significance of the kinematic parameters. (**a**) AP for componentwise parameter. (**b**) AP for different combinations.

Figure 14b compares different combinations of kinematic parameters. We refer to four two-parameter combination bases, the first-order, second-order, translation, rotation parameters and then integrate with other parameters for detection. The AP of the second-order parameter basis (0.48) is significantly higher than that of the first-order parameter basis (0.31). Among the three-parameter combinations, the combination of angular velocity, acceleration and angular acceleration has the highest AP of 0.60. The above results also show significance of the second-order parameters.

## 6. Conclusions

This paper proposes a novel motion-based method MUSAK for drone detection tasks. The biggest difference of the MUSAK method is to use kinematic parameters to describe object motion behavior, and map the retrieved kinematic parameters to three spaces: 3D, pseudo 3D and 2D space for further classification according to quality of the kinematic parameters. Besides achieving the SOTA score on detection precision and recall value, further analyses on the adaptivity and significance of MUSAK also reveal more important facts. Firstly, MUSAK is superior in low-definition scenarios and the 2D motion feature is more robust to image quality degeneration. Secondly, temporal-spatial characteristics demonstrate that the best performance of MUSAK appears at a frame rate close to 25 fps. Thirdly, the second-order kinematic parameters, the translation parameters in the gravity direction and rotation parameters in the direction of the optic axis are of vital importance. These findings may lead to a new direction for detecting moving or blurring objects. In addition, the present work builds a public and more comprehensive drone dataset for future research.

**Author Contributions:** Conceptualization, P.G.; Funding acquisition, G.L.; Investigation, G.L. and P.G; Methodology, S.L., G.L., Y.Z. and P.G.; Software, S.L.; Supervision, G.L., Y.Z. and P.G.; Visualization, S.L.; Writing—original draft, S.L.; Writing—review & editing, Y.Z. and P.G. All authors have read and agreed to the published version of the manuscript.

**Funding:** This research received no external funding.

**Institutional Review Board Statement:** Not applicable.

**Informed Consent Statement:** Not applicable.

**Data Availability Statement:** The self-collected dataset is available online at [45] and other relevant datasets are available in [5].

**Conflicts of Interest:** The authors declare no conflict of interest.

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
