# Peer review of "MUSAK: A Multi-Scale Space Kinematic Method for Drone Detection"

_remotesensing, doi:10.3390/rs14061434_

Round 1

Reviewer 1 Report

The manuscript titled “MUSAK: A Multi-Scale Space Kinematic Method for Drone Detection” represents an outstanding and well prepared scientific paper. The idea of the paper is interesting, and the authors present a novel MUSAK method for drone detection. The manuscript title is accurate and correct. In the entire manuscript, the authors use standard technical and scientific terminology. After an Introduction, the authors explained the Related work, Multi-Scale Space Kinematic Method for Drone Detection and Experiments in detail. Results and Conclusions were conducted according to the scientifically correct approach. The manuscript topics fit in the Remote sensing journal.

I recommend that this manuscript should be accepted after minor revisions.

Comments for authors:

  1. Suggest extending Conclusions with more interesting results and important findings of the research.
  2. Please emphasize more the actual applicability of the proposed paper.
  3. Use MDPI standard font (Palatino Linotype) on figures if you can.
  4. Suggest improving quality of figures, e.g., increase font size on figure 5.
  5. Please, double-check all references and reference styles.
  6. Rename section Conclusion to Conclusions.

Reviewer 2 Report

Overall good analysis of current knowledge in the field.

There are some not well stated and unclear points:
- Row 34 - statement is not true. There are also radar detection methods, radiofrequency detection methoods and many others.
- Confusing paragraph (row 183-193)-  In the previous text, authors said that block scale is corresponding to 3D space while at row 191, they claim that for block scale only the 2D kinematic parameters can be acquired. Need to be explained more deeply.

Reviewer 3 Report

This paper addresses a practical problem of drone detection utilizing three sets of kinematic parameters as input features and gated recurrent units (GRUs) as detectors. The way the authors constructed input features at three scales (detail, edge, and block) is the main novelty, because it is different from the conventional method of down-scaling images and forming the pyramid. Additionally, the manuscript is well-written with high readability. Thus, the reviewer supports its publication after minor revision, where detailed comments are as follows:

  1. Deep neural networks (DNNs) are well-known for their compelling success in many vision tasks, including drone detection. One of the decisive factors behind that success is multi-scale processing capability. Therefore, the statement at lines 39-41 about the limitation of DNNs should be backed with credible evidence.
  2. The authors mentioned that the appearance-based approach (typified by DNNs) worked reliably under complex background and illuminating conditions. However, at lines 49-51, the authors said that the hybrid approach (combination of appearance- and motion-based) only performed well under a relatively clean background. According to the reviewer’s opinion, this statement implies that the hybrid-based approach does not inherit the ability to work well under complex backgrounds from the appearance-based one. Thus, it is highly advisable to double-check the statement at lines 49-51 and make an amendment (if necessary).
  3. The proposed method may not be the first to use kinematic parameters to describe the motion feature of objects. A simple search yields two methods [a,b] that are already published. Hence, please double-check the contributions and revise (if necessary).
    1. Brooks, F. Barbaresco, Y. Ziani, J. -Y. Schneider and C. Adnet, "Drone Recognition by Micro-Doppler and Kinematic," 2020 17th European Radar Conference (EuRAD), 2021, pp. 42-45, doi: 10.1109/EuRAD48048.2021.00022.
    2. X. Seah, Y. H. Lau, and S. Srigrarom, “Multiple Aerial Targets Re-Identification by 2D- and 3D- Kinematics-Based Matching,” Journal of Imaging, vol. 8, no. 2, p. 26, Jan. 2022, doi: 10.3390/jimaging8020026.
  4. It is advisable to analyze the algorithmic complexity and perform a runtime comparison.

Reviewer 4 Report

The author proposes a novel motion-based method MUSAK for the drone detection task, which improves its performance and average precision, and a public and comprehensive UAV dataset has also been created. However, there are still some problems in this paper.

  1. What does SOTA mean in the Abstract, and the authors need to explain the abbreviations that appear for the first time in the paper.
  2. There are many minor writing problems in the whole paper, including Figure1 (c) in line 238, a=1..3 in line 365, and Figure 3 appears twice in the text, the author needs to check the manuscript again.
  3. The formatting of the references needs to be standardized, including the bolding of the year font, etc.

Reviewer 5 Report

The authors present the article entitled “MUSAK: A Multi-Scale Space Kinematic Method for Drone Detection”. The article is interesting and is easy to read. However, it presents the following observations:

line 26 can be justified by considering the following reference: Comparison of PD, PID and sliding-mode position controllers for v-tail quadcopter stability

line 26 can be justified by considering the following reference: A new approach for motor imagery classification based on sorted blind source separation, continuous wavelet transform, and convolutional neural network

All the acronyms should be defined, as GAN.

Add dots at the end of the captions of figs and tables: “Figure 2. An illustration of a ROI”

Line 80: I suggest not using the bold format. This line could be presented with a single sentence.

Please, at the end of the introduction include the structure of the manuscript.

Lines 89-92: I recommend not presenting the references in these lines since they are discussed in the following paragraphs.

Vectorize the figures in order to see the details. Improve the resolution of the pictures.

Add only the reference for the URL of line 440.

Include a nomenclature table.
